# Food Recommendations for Reducing Water Footprint

Ignazio Gallo *,†,‡ , Nicola Landro †,‡ , Riccardo La Grassa † and Andrea Turconi †

Department of Theoretical and Applied Sciences—DISTA, University of Insubria, Via J.H. Dunant, 3, 21100 Varese, Italy; nlandro@uninsubria.it (N.L.); rlagrassa@uninsubria.it (R.L.G.); aturconi@studenti.uninsubria.it (A.T.)
* Correspondence: ignazio.gallo@uninsubria.it
† These authors contributed equally to this work.
‡ Current address: Dipartimento di Scienze Teoriche e Applicate—DISTA, Via O. Rossi, 9—Padiglione Rossi, 21100 Varese, Italy.

**Abstract:** Most existing food-related research efforts focus on recipe retrieval, user preference-based food recommendation, kitchen assistance, or nutritional and caloric estimation of dishes, ignoring personalized and conscious food recommendations resources of the planet. Therefore, in this work, we present a personalized food recommendation scheme, mapping the ingredients to the most resource-friendly dishes on the planet and in particular, selecting recipes that contain ingredients that consume as little water as possible for their production. The system proposed here is able to understand the user's behavior and to suggest tailor-made recipes with lower water quantity used in production. By continuously using the system, the user can gradually reduce their water footprint and benefit from a healthier diet. The proposed recommendation system was compared with the results of two papers available in the literature that represent the state of the art, obtaining similar results. Therefore, the results of the presented recommendation system can be considered reliable.

**Keywords:** sustainable food consumption; food recommendation system; water footprint; food waste; good practice

## 1. Introduction

Water is a critical aspect of the growth and welfare of both humans and the planet since it is a life-giving resource. Nowadays, due to thoughtless and wasteful water management, different issues exist, such as scarcity, lack of drinking water, and pollution. Overall, 70% of the Earth's surface is covered by water. On the other hand, freshwater is limited. This water is needed for drinking, bathing, and farming. As a result, it is critical to attempt to reduce water usage in all areas, including the industrial, food, and household sectors. Therefore, to provide adjustments, offer measurements, and also alert people about actual utilization of water in the production of consumed foods, the water footprint metric has been established by Arjen Hoekstra [1]. The notion of water footprint, defined as the total amount of freshwater utilized to generate goods and services consumed [1], was introduced in order to improve global water management. This value can be calculated for a particular operation, such as rice farming, a pair of shoes, fuel, or for a multinational corporation.

The food industry, according to a recent study provided by Mekonnen and Gerbens-Leenes [2], is one of the most water-intensive industries, as each product requires a particular amount of water to be produced. As a result, in order to decrease water usage, several studies have been conducted to determine the ideal diet in terms of water quantity used in production and people's health. Blas et al. [3] discovered that a diet high in vegetables and legumes and low in red meat reduced the user's impact on the quantity of water used while simultaneously improving his health. As a result, the Mediterranean diet embraces these concepts. Unfortunately, according to a recent study by Blas et colleagues [4], people are shifting their diet away from the Mediterranean and toward the American one. This diet is harmful to both human health and the planet due to the high consumption of meat

and sweets and reduced consumption of vegetables. As a result, the American diet may cause health concerns and increase the impact on the quantity of water used around the planet. In this paper, we present a recommendation system that allows consumers to buy the ingredients they need to make a recipe or buy the ready meals of their choice while reducing their water footprint. All user orders will be taken into account by the recommendation algorithm, which will attempt to understand the user's behaviour, diet and tastes to offer tailor-made recipes or dishes. Furthermore, the system will take into account the water footprint of the recipes to optimize the suggestions provided to the user, always taking into account the reduction of water quantity used in production, as summarized in Figure 1.

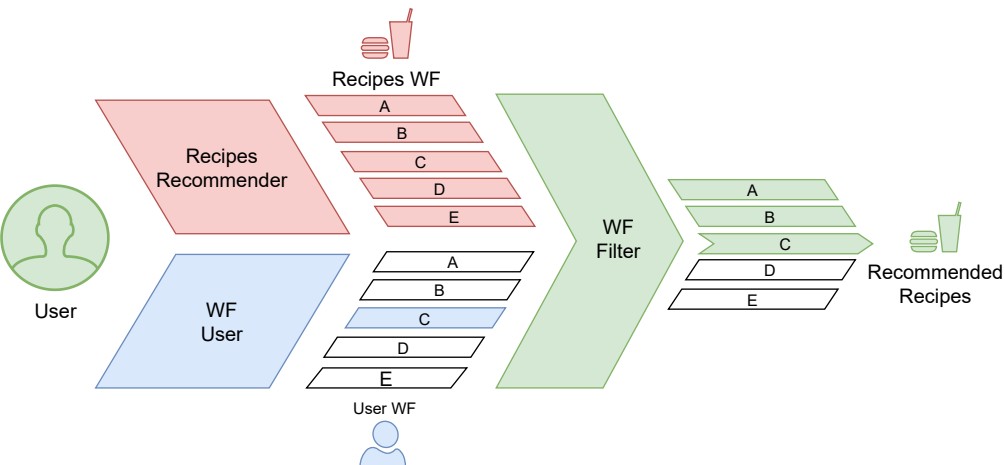

**Figure 1.** Intuitive diagram explaining developed water footprint-based recipe recommendation system. The system must categorize both recipes and users into five different categories (A, B, C, D, E) before recommending water-saving recipes. In the end, the user will only be offered recipes that have a water quantity used in production lower than or equal to the user's category (in the drawing the user's category is C).

To avoid suggesting meals that are opposite to the user's dietary habits but have a lower water footprint, the recommendation system gradually decreases the water quantity used in production of the recipes suggested to the user. It is feasible to avoid recommending dishes to a user that are not suitable for his diet in this way. Simultaneously, recipes that are similar to the user's favourite meals but result in lowering of quantities of water used in production are suggested. Continually recommending dishes with decreased water quantity used in production over time may lead to adjustment of the user's diet. To allow other researchers to start from our approach and conduct new experiments, we release the source code [5] and datasets used [6]. Hence, we divided this paper into seven sections, each of which is summarized below. Section 2 explains the concept of a water footprint, as well as its benefits and applications, and provides an overview of food water quantity used in production. Section 3 explores the literature and provides an overview of current recipe recommendation systems, as well as works that investigate ways to minimize water quantity used in production. Both datasets employed, and the data pre-processing phase, are detailed in Section 4. Section 5 explains the approach proposed in detail, including motivations and explanations for each decision. Section 6 outlines all of the experiments conducted before deciding on the final solution. In addition, it includes all of the results generated by the tests and a brief comparison with state-of-the-art algorithms. Finally, Section 7 summarizes the findings of the study and suggests future research directions.

## 2. Water Footprint

Water scarcity is one of the most pressing environmental issues since water is a life-giving resource for both the planet and humans [7]. As a result, researchers have introduced a new metric called "Water Footprint" to provide measurement and alert

people about water usage [8]. The expression water footprint refers to the influence of human actions on the environment, specifically the use of freshwater associated with the production and disposal of goods and services. There are other parameters to detect humanity's impact on the earth, such as the ecological footprint, which calculates the consumption of natural resources, or the carbon footprint, which measures greenhouse gas emissions made by individuals, businesses, and countries. These indicators are critical for understanding the effects of human actions on Earth, analyzing the environmental effects of the current socioeconomic system, and determining what solutions to implement for a more eco-sustainable future.

Water footprint has a wide range of applications; it can be used to detect the use of freshwater by a corporation, a product, or a city. It is also feasible to calculate the water footprint of an entire state, continent, or the world's total impact on the quantity of water used. The water footprint considers both direct and indirect freshwater quantity used in production. The first is tied to the withdrawal of water, such as drinking or washing clothing, while the second is related to the consumption of water required to obtain a product or service, such as the water necessary to produce a kilogram of beef. The indicator considers both direct and indirect water quantity used in production, as well as water pollution caused by numerous human activities. This provides a comprehensive view of a product's, company's, or country's water footprint, as well as the ability to compare different solutions and discover the most environmentally sustainable options.

The food industry is one of the areas with the highest water quantity used in production; indeed, the production of food necessitates large amounts of freshwater. This resource is critical for cattle farms and agricultural irrigation; however, these regions frequently generate large waste, particularly when enterprises do not embrace sustainable methods and customers make mindful choices. According to the Water Footprint Network (Water Footprint Network. https://waterfootprint.org/, accessed on 12 January 2022), beef has a water footprint of 15,415 L/kg, which is a very high consumption compared to tomatoes (214 L/kg), lettuce (237 L/kg), and everything else in the vegetable category has an average water footprint of roughly 250 L/kg. A more detailed comparison among the various food categories is shown in Figure 2.

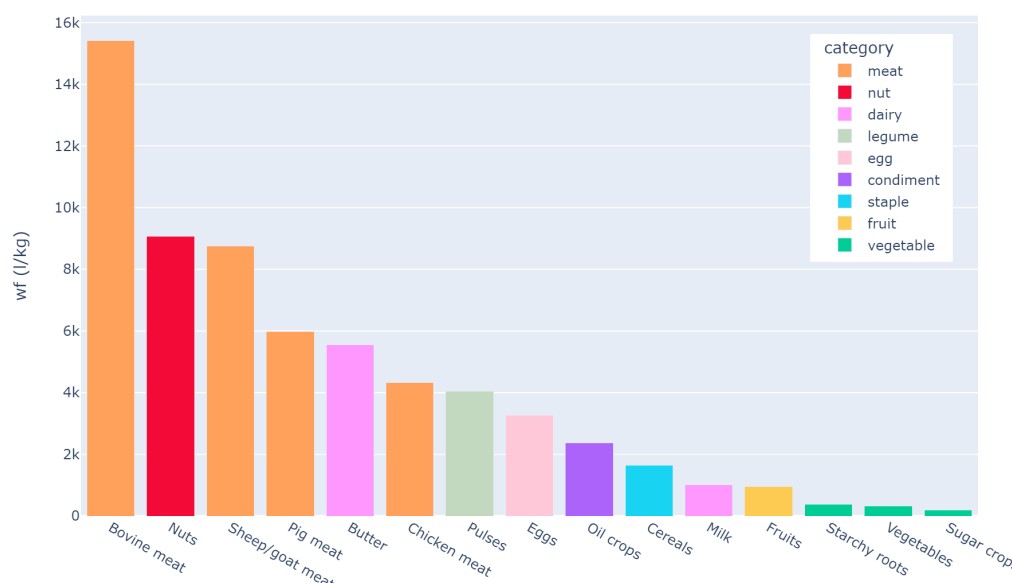

**Figure 2.** The bar plot shows the different categories of food ingredients and their water quantity used in production. The unit of measurement of the water footprint (*WF*) is the liter per kilogram (L/kg).

As shown by these statistics, a healthy diet is also more environmentally sustainable as clarified in the case of Hungary based on a representative dietary survey [9]; for example, fruits, vegetables, and grains have a minimal water footprint. Foods of animal origin, on the other hand, such as meat and cheese, require large use of water resources in order to be produced. To make eco-friendly decisions, starting with shopping, favoring healthier meals and items from short supply chains and local suppliers is the right choice.

In a framework assessing the footprints of food consumption [10] they show how the food demand influences production and, indirectly, the water quantity used in production. That framework makes it possible to assess the environmental impacts of large-scale food consumption patterns and the proposed recommendation system can help to improve the sustainability working on the demand of foods. As the solution proposed in [10] can be used by government to improve the sustainability, the developed system can be used by the food provider helping to act directly on the demand of food.

## 3. Related Works

Food is an essential aspect of human life and health; however, when the food is consumed in excess or incorrectly, it can cause various health problems, so it is important to try to eat a healthy and balanced diet. Following the relevance of food, several studies have been conducted to try to enhance consumer intake and health. Trattner and Elsweiler [11] classified research in food recommendation systems into six categories: content-based techniques, collaborative filtering-based methods, hybrid methods, context-aware approaches, group-based methods, and health-aware methods.

Based on previous actions such as purchases or feedback, a content-based approach suggests components that are most similar to those for which a user has shown a preference. This type of recommender is unable to work without a product description and a complete user profile. This approach is implemented in a variety of ways. For example, Freyne and Berkovsky [12] developed individualized suggestions by analyzing a user's preferences based on specific ingredients; alternatively, Harvey et al. [13] considered not only the ingredients loved but also those that the user dislikes. This method is not restricted to textual data; it can also be enhanced by the use of images since most food-related decisions are performed on the basis of sight. Yang and colleagues [14,15] found that algorithms built to extrapolate crucial visual elements of food photos, such as Convolutional Neural Networks (CNN), can outperform baseline techniques.

Collaborative filtering algorithms recommend items based on an analysis of user preferences, purchases, and behaviors that are common among similar users. Freyne and Berkovsky [12] used Pearson correlation on the rating matrix to evaluate the closest neighbor technique. SVD outperformed both content and collaborative filtering algorithms, according to Harvey et al. [13]. Ge, Elahi, and colleagues [16] offer a matrix factorization technique for food recommender systems that combine rating data with user-supplied tags to produce much higher prediction accuracy than content-based and traditional matrix factorization baselines.

According to Ricci et al. [17] "A hybrid system combining techniques A and B tries to use the advantages of A to fix the disadvantages of B". Regarding the food recommendation challenge, various studies have shown brilliant results: Freyne and Berkovsky [12] used a hybrid technique to incorporate three distinct recommender strategies in a single model, utilizing a switching strategy that targeted groups of users. The switching was determined by the ratio of a user's rated items to the total number of items. Furthermore, Harvey et al. [13], achieved the best results in their studies by combining an SVD approach with user and item biases.

A recommendation system that considers contextual information such as location, social media information, device kind, and many others can be much more successful nowadays. This contextual information is especially significant in the food recommendation; for example, the preparation of a dish, its difficulty, or the many instruments required can make a system more powerful. According to the Harvey et al. [18] analysis, factors

such as how clearly the preparation processes are explained, as well as the nutritional features of the meal, the availability of ingredients, and temporal aspects such as the day of the week have an impact on the user's opinion of the suggestion.

A group recommender system is one that collectively suggests products to a group of users based on their preferences. Using social and behavioral elements of group members to produce group recommendations, in addition to user choices, will improve the quality of the material offered in diverse groups. Despite the fact that this strategy could be very successful for food recommendation tasks, because people do not usually eat or make food choices alone, but are influenced by activities carried out with friends, relatives, or coworkers, research on group-based food recommender systems has been limited. Berkovsky and Freyne [19] test these strategies on real users in a family context. The findings reveal that this type of scenario works well with families; however, it was not possible to produce customized recommendations for all members.

The goal of the health-aware food recommender system is to aid users in making daily diet decisions based on nutrition and health criteria. As a result, a wide range of studies has been conducted in this area. Elsweiler, Hors-Fraile et al. [20]; Elsweiler, Ludwig, Said, Schäfer and Trattner [21]; Schäfer et al. [22] provided a number of research studies that directly include nutritional components into the recommendation technique to enhance the user's health. Instead, Ge, Ricci, and Massimo [23] used a calorie-counting technique in their recommendation system. For most users, Elsweiler, Harvey, Ludwig, and Said [24] discussed the trade-off between proposing what the user wants against what is nutritionally acceptable. The World Health Organization (WHO) (https://www.who.int/, accessed on 22 December 2021) and the United Kingdom Food Standards Agency (FSA) (https://www.food.gov.uk/, accessed on 28 January 2022) provided the nutritional and health information utilized in these publications. Harvey and Elsweiler [25] adopted a different approach, providing an interface that allows the construction of plans that include breakfast, lunch, supper, snacks, and beverages. Substituting ingredients in recipes is another strategy that may be used in order to try to enhance the user's health. In their studies, Achananuparp and Weber [26] and Teng et al. [27] introduced this possibility in order to try to encourage the user to make healthier choices. These solutions, on the other hand, have yet to be adequately evaluated in a nutritional context.

Additionally, water footprint is a burning contemporary problem in terms of sustainability; indeed, one of the centerpieces of the United Nations 2030 Agenda (https://sdgs.un.org/2030agenda, accessed on 14 November 2021) is the responsible management of water resources, with the goal of providing access to sanitation and clean water to the whole population by 2030. In order to accomplish this, it is critical to take action to decrease the water footprint by encouraging eco-sustainable alternatives in the usage of freshwater by individuals and businesses. Numerous research studies on the topic of the water footprint have been published by Alejandro Blas, Alberto Garrido, and Bárbara Willaarts [3,4,28]. They have thoroughly examined the impact on the quantity of water used, food waste, and households' meals in their studies, initially focusing on the Spanish area and then moving on to the European and American ones. In addition to examining these aspects and consumption in these studies, the authors provide remedies and suggestions based on predictive analyses in order to try to minimize consumption and, most importantly, water waste.

In their first study [3], Blas et al., attempted to examine and compare the water footprints of two distinct diets, the American diet and the Italian diet. The outcomes demonstrated that the American diet has a 29% greater water footprint than the Mediterranean diet. Given that a Mediterranean diet is a less water-intensive diet, the findings of this study indicate how it might be helpful to both health and sustainability. As a consequence, it is possible to conclude that adopting a Mediterranean diet would result in decreased water footprint use in both Italy and the United States. As a result, the findings support the hypothesis that low-meat diets are more environmentally sustainable in terms of water conservation, hence assisting in the resolution of the health-environment conflict.

Further study by the same authors [28], focusing on food consumption and waste in Spanish households, demonstrated how essential this problem is for long-term sustainability. The article mainly focuses on an examination of the water footprint of Spanish consumers, concluding that water usage in Spain is equal to 52.933 hm$^3$, approximately 3302 L per person and day. Based on this information, meat, fish, and animal fats (26%), as well as dairy products, make for the majority of the overall water footprint (21%); so, reducing the use of these products in exchange for increased consumption of fruits, vegetables, and legumes would result in significant water savings. Furthermore, the research investigates the impact of food waste on the environment's water footprint. In conclusion, they state that, based on the findings, households might achieve a greater decrease in their food-related water footprint by improving their eating patterns rather than simply avoiding food waste, from a Spanish viewpoint.

A third study, completed recently by Blas et al. [4] examines the Mediterranean diet and compares it to current consumption trends among Spanish citizens. The report claims that current food consumption patterns in Spanish families are trending toward diets that include more meat, dairy, and sugar items, as well as fewer fruits, vegetables, and grains, moving away from the Mediterranean diet. Hence, because it is a less water-intensive diet, following a Mediterranean diet will have health advantages. This assumption is reinforced by the fact that if a Mediterranean diet were to replace current food trends in Spain, the water footprint would be decreased by 753 L per person per day, because the items that consumed the most water were of animal origin: meats, animal fats, and dairy products.

## 4. Dataset

In this study, two different datasets were used for the experiments. The first dataset [6] was provided by PlanEat.eco (https://planeat.eco/, accessed on 30 October 2021), an Italian website that helps consumers reduce food waste, delivering ready ingredients and instructions for cooking recipes. The second dataset was previously made available by [29], collected from a large online recipe website known as Food.com (https://www.food.com/, accessed on 1 October 2021). The two sets of data have a similar structure but differ in the content: the former consists of Italian recipes, written using the Italian language, and user history orders; the latter contains user ratings and reviews scraped from multiple different recipes, which are written in English.

The PlanEat dataset includes all information about recipes, ingredients, quantities, and users orders. It contains a total of 813 recipes, made from approximately 524 ingredients, with each recipe containing 4 ingredients on average. In addition to the recipe data, the dataset also provides information about user history, preferences, and orders. The data span a period of more than 1 year, including a total of 81,627 user orders from March 2020 to July 2021. In the dataset a total of 551 users are available. The second dataset used in the experiments consisted of 180K+ recipes and 700K+ recipe reviews covering 18 years of user interactions and uploads on Food.com. This dataset contains information about recipes along with a raw description, cooking time, tags, nutrition values, steps, and ingredients. It provides a total of 231,637 recipes, made from approximately 14,942 ingredients, with each recipe containing 9 ingredients on average. Furthermore, it also includes a set of 1,132,367 user reviews of recipes from January 2000 to December 2018. User reviews of a single recipe are composed of a textual description, a date, and a rating from 0 to 5. Tables 1 and 2 describe more statistical information and an overview of the two datasets.

**Table 1.** Statistical information about the recipes contained into the two different datasets.

|  | PlanEat.eco | Food.com |
| --- | --- | --- |
| Number of recipes | 813 | 231,637 |
| Number of ingredients | 524 | 14,942 |
| Maximum number of ingredients in a recipe | 21 | 43 |
| Minimum number of ingredients in a recipe | 1 | 1 |
| Average number of ingredients in a recipe | 4 | 9 |

**Table 2.** Statistical information about the orders and reviews contained into the two different datasets.

|  | PlanEat.eco | Food.com |
|---|---|---|
| Number of total orders | 81,627 | 1,132,367 |
| Number of users | 551 | 226,570 |
| Maximum number of orders for a user | 2660 | 7671 |
| Minimum number of orders for a user | 1 | 1 |
| Average number of orders for a user | 148 | 4 |

In addition to the data presented above, we provided a new embedding of ingredients along with their class (e.g., "vegetable" or "meat") and the associated water footprint. All water footprint information in the embedding was measured following the unit of measurement liter on kg (L/kg); for example, bananas have a water footprint of 790 L/kg, which means that 790 L of water are needed to produce 1kg of bananas. From the knowledge of the unit, it is possible to calculate the correct water footprint of a specified ingredient in a recipe based on its quantity. We defined the total water footprint of a single recipe as the sum of the single ingredients' water footprint value, computed on the quantity used in the recipe.

In the dataset, there are more than 800+ ingredients associated with their class and more than 300+ ingredients associated with their water footprint. Water footprint data were collected from the official website of the Water Footprint Network, a platform for collaboration between companies, organizations, and individuals to solve the world's water crises by advancing fair and smart impacts on the quantity of water used. In order to have more complete data on the water footprint, other ingredients, not provided by Water Footprint Network, were then scraped from the HEALabel website (https://healabel.com/, accessed on 28 November 2021). HEALabel is a resource for ethical consumers that provides tons of information about how food, fabric, and brand impact the environment, people's health, and animals. Additionally, the also website contains details of foods' health, sustainability, and water footprint.

In order to have the data, both recipes and reviews, in a suitable form for the proposed model it was necessary to preprocess them. The first step in processing the recipes was identifying the ingredients from the raw text of the recipe. First off all, we removed all the stop words (e.g., "in", "or", "the"), all words that referred to consistency or temperature (e.g., "sliced", "fresh"), cooking methods (e.g., "boiled", "roasted"), tags (e.g., "vegan", "vegetarian"), quantifiers, or numbers (e.g., "10gr", "1unit") and colors (e.g., "red", "green") by regular expression matching. After that, in order to avoid mismatch of similar ingredients such as "Porcini Mushrooms" or "Mushrooms", these types of special ingredients were grouped together in a single form, e.g., "Porcini Mushrooms." and "Mushrooms" became "Mushrooms". Thanks to this preprocessing, both datasets now contain a clean form of the ingredients. Afterward, to avoid memory problems within the proposed model, we decided to keep only the columns that corresponded to the ID of the recipe, its name, its ingredients, and their quantities.

Regarding user orders and reviews, we decided to build the user's profile through his reviews given to the recipes. The Food.com dataset provides the user rating, on a scale from 0 to 5, of single recipes. Instead, unfortunately, the Italian dataset only contains orders placed by users. From these data, we grouped all the orders placed by a single user and assigned a vote to the recipe (always from 0 to 5) considering the frequency and number of times it was purchased. Both the two datasets now contain all orders of users with an associated rating. Accordingly, in order to get good results in training the model, all users and all recipes below a specific threshold had to be removed: based on the information in the previous section, it was decided to delete all users with less than five reviews and all recipes with less than three reviews.

## 5. Proposed Approach

The main goal of the proposed approach was to introduce a recommendation algorithm that takes into account the water footprint of the recipes when suggesting them to the end-user. In order to suggest recipes to the user, it is necessary to know his reviews and his history orders. After understanding his behavior, the algorithm must identify similar recipes that have a lower water quantity used in production. Since the meals with a lower water footprint are vegetarian or those without meat, it is necessary not to recommend to a user, who has a diet with high consumption of meat, totally vegetarian recipes with reduced water quantity used in production. In this way, it is possible to prevent a user from taking the suggestions into account because they are too far from his diet. By continuously recommending recipes similar to those that the user usually orders, but which have a slightly lower water footprint, it is possible to lead him to a different diet than the initial one.

In order to achieve these results, we proposed a novel solution that allowed us to classify recipes into five different categories, from the one with a lower water footprint (*A*) to the higher one (*E*), and assign a score to the user based on the history of his orders. By combining these two classifications within the recommendation system, it is possible to recommend to a user, with a certain score, recipes that belong to certain categories. Finally, we can state that the main difference between the proposed system and a common recommendation algorithm is the possibility of suggesting recipes to users that slightly lower the water quantity used in production. For a better overview of the proposed system [5], it is possible to break it down into three units: a recipe classifier, a user classifier, and a recommendation algorithm. The first unit, explained in Section 5.1, is able to classify a single recipe based on its total water footprint and relate it to all the other recipes in the dataset. Section 5.2 explains the second unit of the proposed system: a user history classifier. To make this second classification, we take into account all the reviews or all the choices made by the individual user and then assign them a score. Finally, Section 5.3 introduces the recommendation algorithm. It takes into consideration the user's score and favorite recipes to recommend the best combinations to reduce the water footprint.

### 5.1. Recipe Classifier

This unit explains the classification process for all the recipes in a category based on their water footprint. To achieve this classification it is first necessary to calculate the water footprint $WF_i$ defined in Equation (1), for every recipe *i* of the dataset. As we have already highlighted in the previous sections, to obtain this value we find the sum of all the results of the proportions obtained between the ingredient quantity in the recipe *i* and its water footprint.

$$WF_i = \sum_{g=1}^{\infty} wf_g \cdot qt_g \tag{1}$$

Hence, we define the formula above (Equation (1)): a recipe's water footprint is equal to the sum of each *g* ingredient's water footprint ($wf_g$) multiplied by the relative quantity $qt_g$. Since the water footprint of individual ingredients is measured in L/kg, all quantities must be converted to kg in order to provide an accurate assessment.

In this way, we get a general figure of how recipes relate to others in terms of water quantity used in production. Therefore, to calculate the water quantity used in production for each recipe, it is necessary to know the exact water footprint of each of its ingredients. For this reason, we made use of the embedding presented in the dataset Section 4 containing several ingredients and their corresponding class and water footprint. Afterward, we ran into another problem: how to correctly match ingredients written in a plural form. To solve this, we tried both lemmatization and stemming. Lemmatization is the process of grouping together the different forms of a word so they can be analyzed as a single item instead; stemming is the process of reducing inflection in words to their root. Lemmatization is similar to stemming but it brings context to the words. Although many research works [30]

state that lemmatization leads to better results than stemming, even for languages other than English, in the conducted experiments we have found that better results are obtained using stemming than lemmatization when working with Italian datasets. Hence, to provide support for different languages, we opted to use stemming. If an ingredient is not present in the embedding, its water footprint becomes equal to the one of its corresponding class, e.g., vegetables, fruit, meat or others. After the calculation, we created a dataset containing recipes and their corresponding water footprint. Recipes were then sorted from the one with the lowest water footprint to the one with the highest water footprint. All the recipes were then divided into five equal parts and the labels *A*, *B*, *C*, *D*, *E* were assigned to each cluster $R_A, R_B, R_C, R_D, R_E$. Intuitively, the recipes that have a water footprint equal to *A* are those that use less water quantity used in production. On the contrary, those with a label equal to *E* exploit a large consumption of water. The main concepts described in this section are summarized in Algorithm 1.

---

**Algorithm 1** Water footprint recommendation system: initialization.

---

**Input:** *d*: complete dataset of ratings

> $U$ is the set of all users
> $F = [A, B, C, D, E]$ is the ordered list of all categories based on water footprint
> $v = \{WF_i | \forall i \in recipes\}$
> $\hat{v} = ascendOrder(v)$
> $R_A, R_B, R_C, R_D, R_E = bucketing(\hat{v})$ , where $recipes = \bigcup\limits_{I \in F} R_I$
> $r_{Iu}$: is the ratings list of all recipes of type *I* selected by user $u \in U$
> $w_I$: is the weight of category $I \in F$
> $n_{Iu} = |r_{Iu}|$
> $X_{Iu} = n_{Iu} \cdot mean(r_{Iu}) \cdot w_I$
> $X_u = \bigcup\limits_{I \in F} \{X_{Iu}\}$
> $\hat{X}_u = \frac{X_u}{max(X_u)}$
> $clusters$ = trainKmeans($k = |F|, features = \hat{X}_u$)
> $I_u$ is the user cluster label $\in F$, obtained with the majority voting on *clusters*

---

*5.2. User History Classifier*

In this section, we describe how to assign a score $u_{score}$ to a single user $u$ based on its history of selected recipes, which therefore involves understanding which recipes, and especially which ingredients, the single user prefers and likes. A score can assume the same labels as the recipes: *A*, *B*, *C*, *D*, and *E*. Thus, if a user's score is equivalent to *E*, it means that the majority of his orders are for dishes in category *E*. Therefore, if the user's score is equal to *E*, it means that the user's diet is fully skewed toward dishes with a huge water footprint. In this recommendation system, assigning a score to users' orders becomes useful for understanding their behavior and diet. Moreover, we employ the score in the system to suggest recipes that are classified into healthier categories. Indeed, the user's score is comparable to the classification of the recipes into categories *A*, *B*, *C*, *D*, and *E* based on their water footprint. Both categories correspond to a scale of values ranging from a small (*A*) to a large (*E*) water footprint.

Unfortunately, because there are no truth values in this classification, it is impossible to properly regulate the algorithm's outcomes. Hence, we adopt clustering to classify users automatically. Afterward, the K-Means [31,32] is fed with the number of reviews carried out for each category of recipes for each user, multiplied by the average value of the same category's ratings. This multiplication is designed to give the user's reviews more weight; in particular, if it has performed a higher order of recipes *A* than recipes *B*, but the average of the ratings of the recipes *B* is higher than the average of the ratings of the recipes *A*, then *B*'s final value may be greater than *A*'s. This is because, despite having placed fewer orders or reviews, the user liked more of the recipes. In addition, multiplication by factors ranging from 0 to 1 is included, which increases the weighting of

the water footprint value of the recipes dependent on their category. For instance, if we choose to add a weight of $w_A = 0.5$ to category $A$, the value of the recipes in this category will be multiplied by 0.5 in this iteration. In the system specified in this work, we decided to leave all categories weight $w_I$ equal to 1, but we also conducted some experiments to see the effect caused by the change in this weight. Finally, these values were then normalized with Min–Max Normalization.

The clustering provides a five-category classification: in category $A$, users have a large number of $A$ recipes and low numbers in the other categories, and so on. In general, we used a majority voting technique to assign a class category to the K-Means result. As a result of this classification, a comprehensive overview of the system's users may be obtained. The main concepts described in this section are summarized in Algorithm 1.

### 5.3. Recommendation System

The main component of the proposed system is described in this section, which deals with the recommendation system algorithm. The approach allows us to combine the findings of the two previous units and suggest recipes to the user that are more similar to his tastes and preferences while also reducing the water consumption of the food production. Collaborative Filtering generated the best outcomes of the other techniques. As a result, the algorithm employed to develop the recommendation system is a K-Nearest Neighbors (KNN) type algorithm. KKNBaseline [33] (from Surprise library (http://surpriselib.com/, accessed on 10 October 2021)), produces the best outcome. This is a basic collaborative filtering algorithm that takes into account a baseline rating. The algorithm determines how $r_{ui}$, or the rating a user $u$ would give to an item $i$ is calculated in the forecasts. The formula of the algorithm is defined as follow (Equation (2)):

$$r_{ui} = b_{ui} + \frac{\sum\limits_{j \in N_u^k(i)} sim(i,j) \cdot (r_{uj} - b_{uj})}{\sum\limits_{j \in N_u^k(i)} sim(i,j)} \tag{2}$$

where $b_{ui}$ is a baseline estimation for an unknown rating $r_{ui}$ and accounts for the user and item effects as reported in the Equation (3).

$$b_{ui} = \mu + b_u + b_i \tag{3}$$

The parameters $b_u$ and $b_i$ indicate the observed deviations of user $u$ and item $i$, respectively, from the average, as reported in [33]. The algorithm is further tuned with the following hyperparameters. Shrunk Pearson correlation ($sim(i,j)$) was employed as the similarity measure, similarities were computed between the recipes instead of the users, and the minimum support number was set to five.

By executing the first portion of the algorithm, it is possible to retrieve all of the recipes that the user could enjoy the most, sorted in descending order. The sorted list of recommendations includes all of the information about the recipe, including its water footprint and associated category. The user's score is then taken into account, and any recipes with a value greater than the user's score are removed from the list. Additionally, to reduce the user's water consumption in a slight and ongoing way, only the recipes with category $A$ and all those with a category two-class-lower than the user's score are maintained. For instance, for a user who has been assigned a score equal to $E$, the system will recommend recipes that belong to categories $A$, $C$, and $D$. As a result, if a user continues to eat and order dishes from the suggestions, the person may be awarded a lower score, such as from $E$ to $D$, in subsequent cycles. Therefore, the recipes that will be recommended to the user belong to categories that are always lower than the user's score. By offering similar recipes to the user and assuming that he picks them from the recommendations, it is feasible to reduce his water consumption. As an outcome, the consumer receives a list of recipes that are ranked by taste but have a smaller water impact. The main concepts described in this section are summarized in Algorithms 1 and 2.

---

**Algorithm 2** Water footprint recommendation system: recommendation.

---

**Input:** $u$: the user, *recipes*: the recipes recommended

   $u_{score} = F.indexOf(I_u)$
   $I_i$: is the class $I \in F$ of recipe $i$
   $i_{score} = F.indexOf(I_i)$
   $WF\_recommended\_recipes = \{i | i \in recipes, i_{score} \leq u_{score}\}$

---

## 6. Experiments and Evaluations

Before deciding on the appropriate algorithm for the recommendation system presented in the previous section, we decided to carry out several experiments to find the approach with the best outcomes. Each algorithm was evaluated and compared to the others using the same dataset and metrics.

In order to select the best methodology, we decided to conduct experiments on two main approaches: content-based and collaborative filtering. In addition, we decided firstly to conduct all evaluations excluding the water footprint filter, to determine the best algorithm capable of understanding the user's behavior and tastes. Furthermore, all experiments are compared to two state-of-the-art approaches in order to determine whether the proposed system's results are acceptable and comparable. Finally, we provide a comparison of the algorithms, including the water footprint filter, to determine whether the loss of user taste is worth the water savings.

The first technique employed to develop the recommendation system is a content-based method. In this approach, only similarities between recipes are taken into account rather than similarities between users. Due to this, it is necessary to analyze the user's behavior, purchases, and reviews to generate his profile. As a result, the system considers the user's profile during the recommendation process to suggest recipes that are most similar to those that he preferred and ordered the most. The similarities between the two recipes are based on the ingredients they both contain. Additionally, each ingredient is assigned a weight in the similarity calculation. This is because condiments such as oil, salt, and pepper are used in many recipes but do not correspond to a critical element in determining the user's preferences. Hence, it is critical to express ingredients and recipes as vectors in order to develop a recipe recommender system. We decided to use Term Frequency Inverse Document Frequency (TF-IDF) on all of the ingredients in the dataset. All of the ingredients had already been processed according to the guidelines outlined in the previous chapter. Thanks to TF-IDF scores, ingredients that are used more frequently in recipes, such as condiments, have a lower weight in the embedding than ingredients that are used less frequently. Afterward, we used the cosine similarity to calculate the similarity between the user profile and the recipes. This metric determines the cosine of the angles between two vectors: the user's profile and the TF-IDF vectors based on the recipe information. This method takes into consideration all of the user's reviews, which are weighted for each rating. Then, it determines which ingredients have received a higher rating from the user, which correlates to a higher user preference. Finally, it searches all of the recipes in the dataset for those that include ingredients that are the most comparable to those discovered earlier. As a result, the algorithm's output is a list of sorted recipes, with the first one being the closest to the user's diet and preferred ingredients.

The second strategy used is a collaborative filtering approach. As described in earlier chapters, only the dataset that matched the user ratings was employed in this strategy. Thus, for this approach, different experiments were carried out using several kinds of algorithms provided by the Surprise library in order to obtain the best results. Hence, the following algorithms were employed: BaselineOnly, SVD, KNNBasic, KNNBaseline, KNNWithZScore, KNNWithMeans, and CoClustering. Additionally, we used GridSearchCV to run these algorithms several times to determine the best hyperparameters. GridSearch is the process of fine-tuning hyperparameters to find the best values for a particular model. Since the value of hyperparameters has a substantial impact on a model's performance, finding the optimal ones is a smart practice. This Scikit-learn library function allows us

to cycle over predefined hyperparameters and fits the model to training data. Finally, the optimal settings may be chosen from the given hyperparameters. To avoid overfitting or selection bias, all of the algorithms discussed so far use the cross-validation approach. Cross-validation was performed three times in distinct subsets of the sample. Afterward, we used the results to calculate the model's prediction performance. Therefore, all of the approaches listed above return a list of recipes that are most similar to the consumers' tastes.

### 6.1. Evaluation Metrics

In traditional Machine Learning projects, there are a variety of metrics that are employed to calculate model accuracy or other errors. For example, accuracy is helpful in classification problems, whereas Mean Average Error (MAE) and Root Mean Square Error (RMSE) are effective in regression problems. Furthermore, other than the Hit Ratio (HR), there are not many additional metrics for recommendation systems. However, because this work employs an offline experimental setup, it focuses on a prediction task and is assessed on the accuracy of the predictions; hence, we decide to employ as evaluation metrics the Root Mean Square Error (RMSE) and the Hit Ratio (HR). The hit ratio in a recommender system is defined as the percentage of users for whom the correct answer is included in an N-item suggestion list. It can also be expressed as HR@n, or Hit Ratio at N. This form specifies a technique for calculating the number of hits in an n-item list of ranking objects. In this work, we employed the Hit Ratio with the Leave-One-Out methodology. With the Leave-One-Out technique, the hit ratio measure is defined in a list of 10 proposed recipes to the user. Then, the most recent review for each user is chosen as the test set, while the remaining is used as training data. Initially, 99 recipes are randomly picked for each user who has never interacted with them. Then these 99 elements are put together with the test item (the actual recipe that the user interacted with). As a result, there are currently a total of 100 entries. Furthermore, we used the recommender system to rank these 100 items according to their estimated probability. The top ten entries were picked from a list of 100. If the test recipe was in the top 10 items, we declared it a hit. After that, the process was repeated for each user. Thus, the Hit Ratio is calculated as the average number of hits.

### 6.2. Results Excluding the Water Footprint Filter

As previously indicated, the first experiments we conducted focused on the algorithm's ability to provide findings that were compatible with the user's diet. As a result, the water footprint filter was not used in these first outcomes, as it may remove the best suggestions for a user. Furthermore, we chose to compare the obtained results with various studies in the literature to determine whether the suggested algorithms were in line with the state of the art of food recommendation systems. We ran all of the algorithms on different datasets to compare and discuss how they performed. Therefore, all of the findings from all of the algorithms configurations provided earlier and executed during the experiments are reported in this section. The two measures mentioned above, RMSE and Hit Ratio, are used to compare the findings. In terms of the RSME, it is worth noting that a lower value signifies better performance. On the other hand, a high Hit Ratio suggests greater results. The algorithms were tested and assessed using the datasets described in Section 4: Planeat's Italian dataset and Food.com's American dataset, respectively. The outcomes of the algorithms' execution are presented in Table 3.

As shown in Table 3, no algorithm achieved the greatest outcomes for both measures. The RMSE for the Content-Based recommendation system cannot be determined since it was not applied to forecast user ratings of recipes. However, it is worth noting that its Hit Ratio number was not excellent, but it ranked among the average results.

**Table 3.** Results on both Planeat and Food.com datasets after removing the water-footprint filter. The results of the algorithms on the Planeat dataset are displayed in the first multicolumn, while the results of the algorithms on the Food.com dataset are displayed in the second multicolumn.

| Algorithm | Planeat Dataset | | Food.com Dataset | |
|---|---|---|---|---|
| | RMSE | Hit Ratio @10 | RMSE | Hit Ratio @10 |
| Content-based with user history | - | 0.16 | - | 0.25 |
| BaselineOnly | 0.8383 | 0.08 | 0.9141 | 0.20 |
| SVD | 0.8523 | 0.18 | **0.9089** | 0.17 |
| KNNBasic | 0.9716 | 0.17 | 1.0354 | 0.03 |
| KNNWithZScore | 0.7433 | 0.03 | 0.9692 | 0.50 |
| KNNWithMeans | **0.7332** | 0.03 | 0.9870 | 0.11 |
| KNNBaseline | 0.8484 | **0.34** | 0.9571 | **0.71** |
| CoClustering | 0.7468 | 0.04 | 1.0404 | 0.01 |

Regarding the Planeat dataset, the KNNWithMeans algorithm produced the lowest error value (0.7332) despite a nearly 0% Hit Ratio (0.04) which means the algorithm correctly predicted the test recipe of only 4% of all users. Therefore, even with the lowest error in predicting user ratings, this system failed to understand additional significant information. Instead, the KNNBaseline algorithm produced the highest Hit Ratio score (0.34). This value indicates that the system correctly predicted the test recipe of 34% of all users. It is conceivable that the result was not optimal, but it is important to remember that using only the user's most recent order as a test recipe does not necessarily define his diet; for instance, this order may be something the user wanted to try even though it is in direct opposition to his diet. When considering the Hit Ratio, this likelihood must also be taken into account. While predicting, the KNNBaseline algorithm achieved an error of 0.8484. Although this value was neither the worst nor the greatest, it was still a good outcome. It is worth noting that the RSME value refers to the error in forecasting the user's vote. In this case, the RMSE was 0.8484, indicating that the model's predictions missed the actual ratings by approximately 0.8484 points. This is not a terrible result on a 0–5 rating scale. As a result, a rating of 3.0 against 3.8 does not make a significant difference in determining whether or not someone loved a recipe. The KNNBasic algorithm for the RSME, the KNNWithZScore method, KNNWithMeans, and the CoClustering for the Hit Ratio were the algorithms that provided the worst results.

On the other hand, considering the American Food.com dataset, the algorithm with the lowest prediction error value was SVD (0.9089). Unfortunately this algorithm was unable to properly understand user behavior, as evidenced by its weak Hit Ratio: barely 17% of all users were accurately predicted. However, as for the Italian dataset, the KNNBaseline method delivered the best results in terms of Hit Ratio (0.71). This value is satisfactory, given that, despite multiple inaccurate and conflicting orders that users may have made, the algorithm accurately predicted the test recipe of 71% of users in the American sample. Given the assumptions mentioned above, it is possible to consider its RSME value of 0.9571 to be a good outcome. Additionally, it is worth noting how the Hit Ratio findings have improved as the dataset's size has grown. This can be caused by a large number of users, which raises the value of the asset. Finally, KNNBasic and CoClustering are the algorithms that achieved the worst outcomes in both measures. Indeed, the respective results in both assessments are inadequate: RMSE 1.0354 and 1.0404, respectively, while for the Hit Ratio only 3% and 2% were corrected on all users.

In conclusion, the KNNBaseline algorithm achieved the best Hit Ratio results on both datasets. Even if it did not rank in first place for RMSE, the results might be considered satisfactory. Since no algorithm was proven to be better, it was decided to employ this method to construct the recommendation system.

*6.3. Comparison with the State of the Art*

In order to understand if the results were in line with the state of the art of referral systems and to understand if the system was genuinely understanding user behavior, we decided to compare these results with other works that had a similar purpose and employed the same dataset. Since Planeat's Italian dataset was only accessible for the development of this elaboration, it wa possible to compare only the American dataset of Food.com. This dataset was built expressly for the research proposed by Bodhisattwa Prasad Majumder and colleagues, which involved the production of personalized recipes based on a user's preferences. As a result, their approach was completely different from that described in this paper, and a comparison of the outcomes is impossible.

However, two other papers that used the same dataset employed a methodology similar to the one defined in this recommendation system. Knowledge Graph-Based Recipe Recommendation System, a study proposed by Ricardo Manuel Gonçalves da Silva [34], describes the implementation, development, and outcomes of a recommendation system in the culinary industry. The assessments of the algorithms utilized in this research are based on the same Food.com dataset. In addition, the findings of algorithms from the Surprise library are provided, with some of them being included in the suggested recommendation system's experiments. The author proposes a Knowledge Graph-based approach to developing a recommendation system in this research. He also compares the findings of the BaselineOnly, CoClustering, SVD, and KNNWithZScore algorithms from the Surprise library, which are presented in the form of RMSE. Therefore, the results of Ricardo Manuel Gonçalves da Silva's article may be compared to the outcomes of this thesis work to determine if the recommendation system is in line with the current state of the art.

Table 4 illustrates the outcomes of the algorithms developed in this work, in the center column, and in the work [34], in the right column. The KNNBaseline algorithm, which gave the best results in terms of Hit Ratio in the recommendation system proposed in this chapter, was not employed in the comparative article. Furthermore, it is worth noticing that the algorithms' results were very close to each other. The best RMSE ratio was obtained with the SVD approach in both works, with values of 0.9089 in this study and 0.9203 in the publication [34]. The difference in RMSE between the identical algorithms is due to differences in how the algorithms are configured in the different studies. In the case of BaselineOnly, SVD, and KNNWithZScore, this can lead to better results, but in the case of CoClustering, it can lead to worse results. To summarize this comparison, the SVD algorithm produced the best RMSE values in both works, whereas the CoClustering technique produced the worst. As a consequence, based on this assumption, it is reasonable to conclude that the proposed recommendation system is consistent with the findings available in the state of the art.

**Table 4.** Algorithm results compared with [34] on Food.com dataset.

| Algorithm | Our RMSE | [34] RMSE |
|---|---|---|
| BaselineOnly | 0.9141 | 0.9310 |
| SVD | **0.9089** | **0.9203** |
| KNNWithZScore | 0.9692 | 0.9973 |
| KNNBaseline | 0.9571 | - |
| CoClustering | 1.0404 | 1.0112 |

Furthermore, a comparison with another paper that employs the same dataset is feasible. This work, led by Yinghao Sun and Helena Huang [35], proposes a hybrid approach for a recipe recommendation system that includes the addition of textual information tailored to the user's preferences. Two primary approaches are employed: the first is collaborative filtering, which mostly depends on user ratings of recipes, and the second is a content-based method, which primarily focuses on recipe characteristics. The RMSE

is also used to analyze the results of the recommendation system in this study. In order to prevent memory errors, the authors opted to perform the experiments with only 8% of the dataset to assess their system. The data used to train the model were then randomly selected from the trainset, then the model was subsequently tested on the validation dataset. Hence, in order to deal with this publication, we followed the provided instructions for creating the same training and validation dataset. The implemented algorithms was then run and assessed using these data. The study employs a variety of algorithms for various experiments, two of which are the most similar to those employed in this paper.

The comparison of the various outcomes and experiments is shown in Table 5. The algorithms employed in paper [35] are: the Matrix factorization, which is equivalent to the SVD technique, and a KNN approach, which is comparable to the one used in this thesis' recommendation system. As can be observed from the RMSE values, the results were similar for both algorithms; however, the SVD technique shown in paper [35] was more efficient, whereas the KNN strategy presented in the proposed system was slightly better.

**Table 5.** Algorithm results compared with [35] on Food.com dataset.

| Algorithm | Proposed RMSE | [35] work RMSE |
|---|---|---|
| Matrix factorization (SVD) | 1.3360 | 1.2763 |
| KNN (KNNBaseline) | 1.3355 | 1.3469 |

In conclusion, it is possible to state the findings of the algorithms reported in this report are comparable to those found in the current state of the art. No algorithm has been able to overcome the other results, but all of them are pretty similar. As a consequence, the results generated by the KNNBaseline algorithm in the recommendation system are reliable.

*6.4. Results Considering the Water Footprint Filter*

While considering the water footprint, the RMSE statistic is not suitable for assessing the quality of the proposed algorithm. Thus, to evaluate the pproposed system, we decided to compare the algorithms based on their relative Hit Ratio and the presence or absence of the water footprint filter. Making use of this evaluation as a guide, it is feasible to determine if it is worth decreasing diet-only suggestions to minimize the user's water footprint.

In order to assess the algorithms on the defined metric, we chose to employ the average water footprint value derived from the suggestions for each user in the dataset as our statistic. To make things clearer, this value was calculated as follows: first, the algorithm was run for each user in the dataset, and the sum of the water footprints of the top 10 recipes suggested to each user was calculated. The final value we used for comparison, referred to as average WF in the tables, was the average of all the WF sums for each user.

Hence, in order to check if the water footprint reduction algorithm works properly, we employed both datasets in the evaluation. The outcomes of the several methods mentioned above may be seen in the two tables below. The table has four columns, except for the first, which lists the names of the algorithms. The Hit Ratio value of the algorithm under examination and the Average WF value without employing the water footprint filter are considered in the second and third columns, respectively. Instead, the Hit Ratio and Average WF values derived by the algorithm using the water footprint filter are represented in the last two columns.

The findings using the Italian Planeat.eco dataset are shown in Table 6. From this table, it is clear that the KNNBaseline algorithm produces the best Hit Ratio results, regardless of whether the filter is employed or not. The Hit Ratio value, in the system configuration that considers the water footprint filter, is lower than the previous one, since several recipes that the user may include in his diet have been removed due to their high water footprint. This assertion is backed up by the respective findings in the *Average WF* column: in the first

scenario, without the filter, the water footprint value of recipes suggested to users is around 4500 L; however, employing the filter, the value of *Average WF* is reduced by roughly 50%.

**Table 6.** Algorithm results on Planeat dataset.

| Algorithm | Planeat Dataset | | | |
| | Hit Ratio @10 No Filter WF | Average WF No Filter WF | Hit Ratio @10 Filter WF | Average WF Filter WF |
| --- | --- | --- | --- | --- |
| BaselineOnly | 0.08 | 4303.60 | 0.07 | 2152.47 |
| SVD | 0.17 | 4310.08 | 0.14 | 2234.86 |
| KNNBasic | 0.17 | 4937.72 | 0.11 | 2557.13 |
| KNNWithMeans | 0.05 | 4620.19 | 0.05 | 2371.53 |
| KNNWithZScore | 0.03 | 4656.82 | 0.03 | 2265.78 |
| KNNBaseline | **0.33** | 4904.95 | **0.21** | 2485.22 |
| CoClustering | 0.06 | 4712.97 | 0.06 | 2501.87 |

On the other hand, Table 7 shows the findings achieved utilizing the Food.com dataset. The KNNBaseline method, as can also be seen in this table, produced the best outcomes in both system settings. Additionally, in the system configuration that employed the water footprint filter, two algorithms (KNNBasic and CoClustering) enhanced their Hit Ratio value. However, compared to the system without filtering, the Hit Ratio values for the remaining algorithms decreased. Nevertheless, the Hit Ratio's value dropped as a result of a substantial reduction in the *Average WF* value. As the table depicts, there was a significant difference in these values between the two systems. Using the water footprint filter, the system was able to reduce the average water footprint value by roughly 52%, going from around 23,000 L to around 10,600 L. This difference also emphasizes the high water consumption in the diet of American consumers.

**Table 7.** Algorithm results on Food.com dataset.

| Algorithm | Food.com dataset | | | |
| | Hit Ratio @10 No Filter WF | Average WF No Filter WF | Hit Ratio @10 Filter WF | Average WF Filter WF |
| --- | --- | --- | --- | --- |
| BaselineOnly | 0.20 | 22,597.40 | 0.19 | 10,610.43 |
| SVD | 0.17 | 22,775.64 | 0.17 | 10,628.85 |
| KNNBasic | 0.03 | 22,422.43 | 0.08 | 10,523.25 |
| KNNWithMeans | 0.11 | 23,226.38 | 0.15 | 10,754.47 |
| KNNWithZScore | 0.50 | 23,526.11 | 0.37 | 10,993.42 |
| KNNBaseline | 0.71 | 22,792.02 | 0.51 | 10,879.17 |
| CoClustering | 0.01 | 22,787.95 | 0.04 | 10,694.28 |

Comparing the findings of the two tables, we can state that the recipe filter reduces the water footprint fairly efficiently in both datasets. Unfortunately, as a result of this conclusion, the Hit Ratio value of the algorithms is reduced. However, as the data show, this reduction is characterized by a high level of water savings within the users' suggestions. As a result, the few scores lost between the two Hit Ratios might be minimal when compared to the large difference in water consumption. Additionally, the algorithm might be tuned with different settings to minimize the water footprint filter's resilience while increasing the Hit Ratio value.

To better understand how the algorithm proposed in Algorithms 1 and 2 works, we perform other experiments as the parameter $w_I$ varies. The two plots of Figures 3 and 4 show how the content of the clusters generated by K-means changes as the weight $w_I$ we have proposed changes.

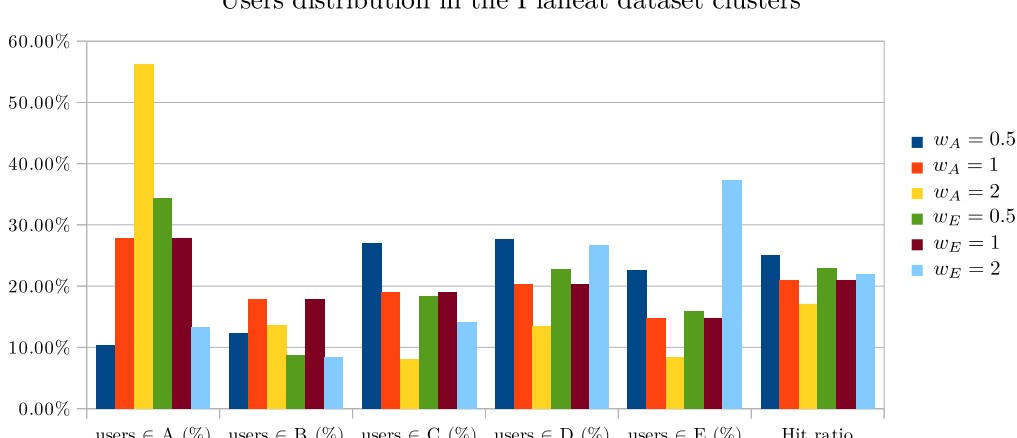

**Figure 3.** The plot shows how the content of the clusters, generated by the K-means on Planeat dataset, changes as the weight $w_I$ we have proposed changes. Furthermore, we compute the hit ratio (HT@10), obtained using the algorithm that produced the best results using the *WF* filter.

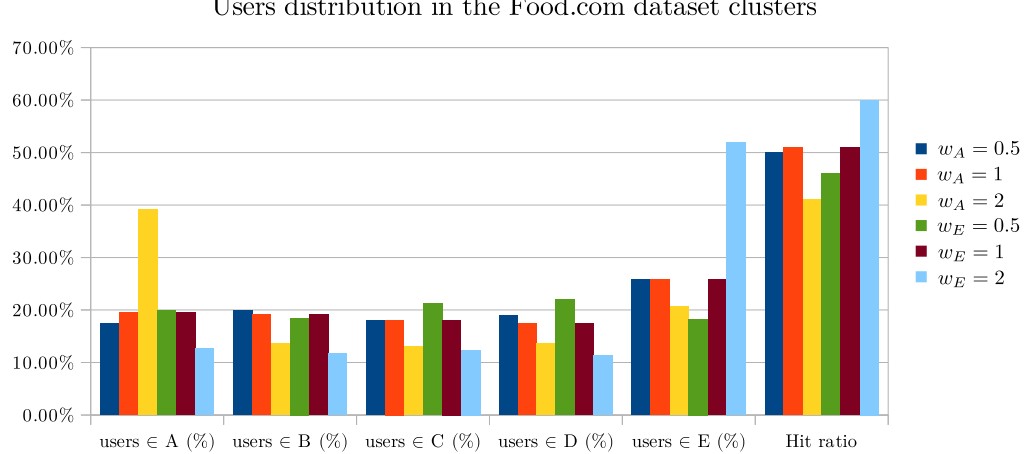

**Figure 4.** The plot shows how the content of the clusters, generated by the K-means on Food.com dataset, changes as the weight $w_I$ we have proposed changes. Furthermore, we compute the hit ratio (HT@10), obtained using the algorithm that produced the best results using the *WF* filter.

Analyzing the results on the Planeat dataset we can see that by setting $w_A = 0.5$ or $w_E = 0.5$ the Hit ratio generally improves. Instead, applying the same values of $w_I$ to the food.com dataset, the Hit ratio increases only with $w_E = 2$, most likely because American users have a higher percentage of recipes with a high water footprint while Italians have a lower percentage in general.

In conclusion, we can state that the algorithm's outcomes without considering the water footprint filter are comparable to the state of the art. This behavior denotes the ability of the system to suggest dishes that are similar to the user's diet. However, whether it is important to minimize the water footprint in order to preserve the planet and human health, each individual's diet must be adjusted, starting with the consumption of recipes and ingredients with a smaller water footprint. Thus, the proposed methodology guarantees a significant reduction of the water footprint in tailored recommendations for users at the expense of a little dietary change.

### 7. Conclusions

Nutrition, as previously stated, is a vital aspect of people's health; additionally, when the notion of water footprint is added, it becomes even more crucial to the planet's sustainability. Indeed, a Mediterranean-style diet rich in vegetables and vitamins benefits not just people's health but also the health of the planet, owing to the consumption of less high-water-consumption ingredients such as bovine products. However, in recent years, consumers' diets have shifted negatively, favoring meat and sweets at the expense of vegetables and legumes. Therefore, each person's overall objective should be to reduce water waste and follow a diet that reduces water consumption. This aim, however, is extremely difficult to fulfill because the benefits are not immediately noticeable and a significant amount of effort is required. The final result of the paper suggests the Mediterranean diet as the reference diet, which includes fruit, vegetables, fish and little meat. Even if the system takes into account the consumption of fish and seafood, among these categories we have chosen to give greater importance to fruit and vegetables not only because they are often considered secondary ingredients but also in relation to the recent new trends towards vegetarian and vegan cuisine so as to also meet the new needs of users.

In conclusion, we can state that the proposed system is capable of understanding the user's behavior and suggesting tailored recipes with lower water consumption. By continually using the system, the user can gradually reduce his water consumption and benefit from a more healthy diet. Additionally, we compared the recommendation system with two publications in the literature to see whether the findings provided were similar to those offered by state-of-the-art algorithms. As we have previously said, the proposed algorithms provide similar outcomes to those produced by state-of-the-art works; hence the results of the presented recommendation system may be considered reliable. Furthermore, the proposed algorithm is also configurable in order to better adapt it to the dietary characteristics of a particular population.

An additional future scenario is the online evaluation of the suggested recommendation system. Rather than utilizing metrics, the system will be assessed based on the votes of actual users who utilize the system. This technique may result in a more accurate evaluation of the system from the end-user's perspective. Since the consumption of fish could be a valid alternative to meat in terms of water footprint, in a future development of this work we want to deepen the comparison between a diet rich in meat and one rich in seafood and fish.

**Author Contributions:** I.G., N.L., R.L.G. and A.T. have contributed equally to the realization of this paper. All authors have read and agreed to the published version of the manuscript.

**Funding:** This research received no external funding.

**Institutional Review Board Statement:** Not applicable.

**Informed Consent Statement:** Not applicable.

**Data Availability Statement:** We released the source code [5] of the proposed algorithm and the datasets used in the experiments [6] in publicly available repositories.

**Acknowledgments:** We are grateful to the Planeat (https://planeat.eco/, accessed on 20 March 2022) web platform for providing us with their data useful for carrying out experiments on the proposed system.

**Conflicts of Interest:** The authors declare no conflicts of interest.

**Publishing Ethics:** The manuscript is our own original work and does not duplicate any other previously published work. The manuscript has been submitted only to the journal *Global Sustainability* and it is not under consideration, accepted for publication, or in press elsewhere. All listed authors know of and have agreed to the manuscript being submitted to the journal. The manuscript contains nothing that is abusive, defamatory, fraudulent, illegal, libelous, or obscene.

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
