# Peer review of "Food Recommendations for Reducing Water Footprint"

_sustainability, doi:10.3390/su14073833_

Round 1

Reviewer 1 Report

The Paper is quite up to the mark but it lacks certain grammatical mistakes like missing some punctuation. The paper is quite structured and below said corrections can be implemented for betterment:

Q1. The more latest literature review can be added in the introduction part including their proper references.

Q2. The Paper can be accepted although some sentences can be shortened and grammatical mistakes can be removed.

Q3. More references can be added to support the result and discussion part in every aspect of influencing factors. Detailed comments are attached below.

 sustainability-1615960

Manuscript: " Food Recommendation for Reducing Water Footprint

Comments: 

There are a few grammatical and punctuation mistakes in the manuscript which need to be correct

  1. ed. Besides, the following correction should be incorporated: -

General Comments. 

  1. The document, when analyzed on Plagiarisms software, i.e., Turnitin, is showing 18%. As per my view, it must be lower down up to 14% or so for an article. Self-Plagiarisms are also not accepted more than 2.5%.
  2. The tables and figures used are not clear and can be enhanced. Heading must be with sequential numbers like 1.0, 1, 1.2, etc. 
  3. In reference section some reference is de-shaped, may be due to formatting. They are also needed to be corrected as per journal format.
  4. The introduction must be reduced to one and a half pages.
  5. The title needed significant modification.
  6. The numbering of content must correct.
  7. The manuscript requires an extension of the literature.
  8. The manuscript does not illustrate great attention and activity in the field.
  9. Tables also contain few references.
  10. Please enhance the manuscript on analysis of earlier mention issues.
  11. The figure number is distorted and can be rechecked.
  12. For the text clarity, would you refrain from using additional words, mostly meaningless filler words, which can be omitted or some archaic words see, e.g. "respectively", "thus", "hence", “therefore", "furthermore", "thereby", "basically,", "meanwhile", "wherein", "herein", "Nonetheless", "Perceivably," etc.?

Specific Comments.  

  • Introduction: The authors should describe the importance of their research more clearly. The references cited lack articles on emerging contaminants from last year. So, add more references (2014-2021) to support the author's points of view.

Abstract: It is needed to be started with small introduction and then quantitative description of the paper. It is also suggestive to shorten few unnecessary sentences in abstract. Underscore the scientific value-added to your paper in your abstract. Please look at articles we have published for models. Your abstract should clearly state the essence of the problem you are addressing, what you did and what you found and recommend. That would help a prospective reader of the abstract to decide if they wish to read the entire article.

1.0 Introduction: Last paragraph must be an outline of the complete study showing the needed and targets assumed in the paper. Hence need minor revision. It also suggestive to add latest article in references. Please use the literature background on cleaner production/sustainability (but not self-citations, please) to broaden the manuscript foundation. Please develop a better title. This one does not state what is important to catch the prospective reader's attention.

  1. Water footprint: More specific details are needed to be added with use of latest reference
  2. Related works: Better to discuss main and important points in tabular form. Table 1 and Table 2 need to be more elaborative and detailed.
  3. Proposed approach: This needed to be shortened and more reader friendly. Use diagrams or table for the same.
  4. Results and discussion: Discussion and conclusions clearly do not establish a strong correlation with cleaner production/sustainability/environmental concerns (however, as much as possible, avoid self-citations). In your discussion section, please link your empirical results with a broader and deeper literature review. Discussions and conclusions must go deeper, it would be more interesting if the authors focus more on the significance of their findings regarding the importance of the interrelationship between the obtained results and sustainable development/cleaner production in the sector context, and the barriers to do it, what would be the consequences, in the real world, in changing the observed situation, what would be the ways, in the real world, to change/improve the observed situation.

 Why and how the said parameters were selected for this work. More specific details are needed to be added with use of latest reference. Use of some pictorial; diagram will be more elaborative for readers.

Figure 1 and Figure 2,3,7 need major revision w.r.t to clarity. They also need more discussion w.r.t to text.

Future scope of this study can be added as well as social impact can also be discussed in this paper.

Conclusions: This section is needed to be free from any variables are symbols. Only main pointed like what was expected and what was achieved must be written. What signification contribution this study to the society must be mentioned in this section.  Please make sure your conclusions' section underscores the scientific value-added of your paper and/or the applicability of your results. Highlight the novelty of your study. Clearly discuss what the previous studies that you are referring to are. What are the Research Gaps/Contributions? In your conclusions, please discuss the implications of your research.

Author Response

Thanks for the comments and suggestions that lead to an improvement of our paper.
Since many of your suggestions also match the requests made by other reviewers, we have already satisfied many of your requests.
We fixed many sentences as you suggested and correct some grammars mistakes. We also improve the references by adding some more recent papers.

Reviewer 2 Report

Pg 4, L108: The authors should elaborate more on the ‘incorrect way of food is being consumed’

Pg 4, L113: Provide a table summarizing these food recommendation systems

In conclusion, the authors suggest people to follow a diet that reduce water consumption, however, it is important to take note that water is essential to human health.  Up to 60% of human adult body is water. The human body can last weeks without food, but only days without water.

Author Response

Thanks for the comments and suggestions that lead to an improvement of our paper.

We report the answers after the questions

Q1: Pg 4, L108: The authors should elaborate more on the ‘incorrect way of food is being consumed’
A1> We elaborated the paper as you suggested.

Q2: Pg 4, L113: Provide a table summarizing these food recommendation systems
A2> In Fig.1 we show a very synthetic overview of the proposed algorithm and, in Alg.1, Alg.2 we report a very detailed summary in terms of pseudocode of the proposed algorithm.

Reviewer 3 Report

It was my pleasure to review research paper Food Recommendation for Reducing Water Footprint. The manuscript deals with the possibility to reduce water usage in food production chain through development and utilization of food recommendation algorithm which incorporates both, consumers’ preferences towards certain dietary patterns and data regarding water footprint in production of different food types. In the manuscript the concept of water footprint with the overview of water usage in production of different foods is explained at one, and thorough insight into development of food recommendation systems is provided on the other hand. Starting from these two concepts the manuscript deals with development and validation of algorithms for food recipes recommendation, while taking into accounts the historic records of consumer’s preference and aiming at selection of combination of meal ingredients with minimal water footprint. The results point out at moderate success regarding the matching of recommendations with consumers’ preferences but point out that highly significant reduction of water footprint can be achieved through application of suitable recipe recommendation system.

However, there are some points in the manuscript which can be improved, and some which have to be corrected.

In spite of great, systematic and complete explanation of developed concept and its effects in the results the quality of the manuscript is diminished due to introduction in which a number of omissions was noted as indicated with the comments in the text and listed here:

Line 20: water footprint is not only about the water waste but rather about quantities of water needed and used in different production steps in food production chain

Line 22: instead of „this usage“ I suggest more clear expression, for example „actual utilization of water in production of consumed foods“

Figure should be moved after the text in which its explanation is provided. For example, after line 46.

Try to depersonalize the manuscript wherever possible. For example in title of figure 1 omit word „our“, rather use „developed“ or simply avoid this word. Similarly depersonalize the text throughout the manuscript.

Line 30, again, you should speak in general about quantities of used water, not only about the wasted water

Line 31: the word „consumption“ in used sentence construction may confusingly refer to consumption (drinking) of water by the consumer. Thus, avoid using water consumption, use other wording like for example „water quantity used in production“ „water footprint“ or something like that.

Line 32: Instead of „water use“, which is again confusing I suggest „impact on quantity of water used“.

Lines 33-38. In my opinion identification of diet in which ingredients with lower water footprint are used with Mediterranean diet, and oppositely the one with high water footprint with American diet is an issue which deserves much deeper investigation with other aim than the one stated in this manuscript. Thus, I recommend that you avoid here, as well as at other places through the manuscript where such comparisons were used the statements about diet types. The manuscript is strong enough without such comparisons which could be without more extensive referencing or results be questioned. If you stick to such statements please justify them with appropriate references.

Line 47: instead of „diet“ I suggest „dietary habits“.

Line 51: instead of „require less water“, I suggest „result in lowering of quantities of water used in production“

Lines 53-54: In accordance with abovementioned I suggest that the sentence „As a result, this recommendation system aims to guide the user toward Mediterranean-style diet“, should be deleted.

Line 98: instead of „family“, I suggest „category“

Line 99: Use „among“, instead of „between“

Line 100-103: the statement that healthy diet is also more environmentally sustainable needs reference based justification

Figure 2- title: instead of „world cuisine“, use „food ingredients“. Also the explanation „The unit of measurement of the water footprint (WF) is the liter per kilogram (l/kg). All data on water use, shown in the plot, was calculated using 1 kilogram of product.“ Is not needed, it is obvious from the plot

Lines 177 and 183: instead of „heated“, use „burning“

Lines 189-190: provide reference numbers here for Alejandro Blas, Alberto Garrido, and Bárbara 190 Willaarts

Line 367: should it instead of (B) stand (E) for large water footprint?

Line 393: The part “reducing his water consumption“, should be rephrased, it could be understood that water consumption during the meal is reduced

I recommend that introduction should be checked regarding English by the professional. In their parts of the manuscript, as far as I see, English is OK.

In my opinion this is an excellent manuscript systematically presenting development and validation of an excellent and, from the standpoint of sustainable food production, very important idea.

Author Response

Thanks for the comments and suggestions that lead to an improvement of our paper.
We report the answers after the questions

Q1: Line 20: water footprint is not only about the water waste but rather about quantities of water needed and used in different production steps in food production chain
A1> As you suggested, we have changed the concepts of "waste" to those of "usage"

Q2: Line 22: instead of „this usage“ I suggest more clear expression, for example „actual utilization of water in production of consumed foods“
A2> We changed the sentence following your suggestions. 

Q3: Figure should be moved after the text in which its explanation is provided. For example, after line 46.
A3> We placed the figures immediately after the place were they are referenced but we do not guarantee that our modification will be accepted by the editors. This is because the latex template we used could position the figures according to rules that cannot be directly modified.

Q4: Try to depersonalize the manuscript wherever possible. For example in title of figure 1 omit word „our“, rather use „developed“ or simply avoid this word. Similarly depersonalize the text throughout the manuscript.
A4> We have depersonalized most of the sentences as you suggested. 

Q5: Line 30, again, you should speak in general about quantities of used water, not only about the wasted water
A5> We changed the text as you suggested.

Q6: Line 31: the word „consumption“ in used sentence construction may confusingly refer to consumption (drinking) of water by the consumer. Thus, avoid using water consumption, use other wording like for example „water quantity used in production“ „water footprint“ or something like that.
A6> We applied your suggestions in all the paper.

Q7: Line 32: Instead of „water use“, which is again confusing I suggest „impact on quantity of water used“.
A7> We applied your suggestions in all the paper.

Q8: Lines 33-38. In my opinion identification of diet in which ingredients with lower water footprint are used with Mediterranean diet, and oppositely the one with high water footprint with American diet is an issue which deserves much deeper investigation with other aim than the one stated in this manuscript. Thus, I recommend that you avoid here, as well as at other places through the manuscript where such comparisons were used the statements about diet types. The manuscript is strong enough without such comparisons which could be without more extensive referencing or results be questioned. If you stick to such statements please justify them with appropriate references.
A8> We cited a recent study by Blas et all[4] to support the statements about diet type.

Q9: Line 47: instead of „diet“ I suggest „dietary habits“.
A9> done

Q10: Line 51: instead of „require less water“, I suggest „result in lowering of quantities of water used in production“
A10> done

Q11: Lines 53-54: In accordance with abovementioned I suggest that the sentence „As a result, this recommendation system aims to guide the user toward Mediterranean-style diet“, should be deleted.
A11> done

Q12: Line 98: instead of „family“, I suggest „category“
A12> done

Q13: Line 99: Use „among“, instead of „between“
A13> done

Q14: Line 100-103: the statement that healthy diet is also more environmentally sustainable needs reference based justification
A14> We added citation [9] to support the sentence.

Q15: Figure 2- title: instead of „world cuisine“, use „food ingredients“. Also the explanation „The unit of measurement of the water footprint (WF) is the liter per kilogram (l/kg). All data on water use, shown in the plot, was calculated using 1 kilogram of product.“ Is not needed, it is obvious from the plot
A15> done

Q16: Lines 177 and 183: instead of „heated“, use „burning“
A16> done

Q17: Lines 189-190: provide reference numbers here for Alejandro Blas, Alberto Garrido, and Bárbara 190 Willaarts
A17> Done

Q18: Line 367: should it instead of (B) stand (E) for large water footprint?
A18> Yes

Q19: Line 393: The part “reducing his water consumption“, should be rephrased, it could be understood that water consumption during the meal is reduced
A19> We changed in "the water consumption of the food production".

Q20: I recommend that introduction should be checked regarding English by the professional. In their parts of the manuscript, as far as I see, English is OK.
A20> We have tried to improve English in the introduction.

Round 2

Reviewer 1 Report

The said correction were done thereby improving the overall overview of current manuscript.Hence can be considered for future publications.

Author Response

Thank you for accepting our changes/improvements made to the paper.

Reviewer 3 Report

Authors improved manuscript taking into account all recommendations provided by the reviewer. The manuscript can be published in present form.

Author Response

(The authors gave the same response as above.)
